# Theoretical Analysis and Expression Profiling of 17β-Hydroxysteroid Dehydrogenase Genes in Gonadal Development and Steroidogenesis of Leopard Coral Grouper (*Plectropomus leopardus*)

**DOI:** 10.3390/ijms25042180

**Published:** 2024-02-11

**Authors:** Mingjian Liu, Hui Ding, Chaofan Jin, Mingyi Wang, Peiyu Li, Zhenmin Bao, Bo Wang, Jingjie Hu

**Affiliations:** 1MOE Key Laboratory of Marine Genetics and Breeding, College of Marine Life Sciences, Ocean University of China, Qingdao 266003, China; 15222387853@139.com (M.L.); ding425926225@163.com (H.D.); jinchaofan@ouc.edu.cn (C.J.); mingyi9802@163.com (M.W.); 17860843380@163.com (P.L.); zmbao@ouc.edu.cn (Z.B.); 2Key Laboratory of Tropical Aquatic Germplasm of Hainan Province, Sanya Oceanographic Institution, Ocean University of China, Sanya 572025, China; 3Hainan Seed Industry Laboratory, Sanya 572025, China; 4Southern Marine Science and Engineer Guangdong Laboratory, Guangzhou 511458, China

**Keywords:** *Plectropomus leopardus*, 17β-Hydroxysteroid Dehydrogenase, sex differentiation, steroidogenesis, gonadal development

## Abstract

The differentiation and developmental trajectory of fish gonads, significantly important for fish breeding, culture, and production, has long been a focal point in the fields of fish genetics and developmental biology. However, the mechanism of gonadal differentiation in leopard coral grouper (*Plectropomus leopardus*) remains unclear. This study investigates the 17β-Hydroxysteroid Dehydrogenase (Hsd17b) gene family in *P. leopardus*, with a focus on gene characterization, expression profiling, and functional analysis. The results reveal that the *P. leopardus*’s Hsd17b gene family comprises 11 members, all belonging to the SDR superfamily. The amino acid similarity is only 12.96%, but conserved motifs, such as TGxxxGxG and S-Y-K, are present in these genes. *Hsd17b12a* and *Hsd17b12b* are unique homologs in fish, and chromosomal localization has confirmed that they are not derived from different transcripts of the same gene, but rather are two independent genes. The Hsd17b family genes, predominantly expressed in the liver, heart, gills, kidneys, and gonads, are involved in synthesizing or metabolizing sex steroid hormones and neurotransmitters, with their expression patterns during gonadal development categorized into three distinct categories. Notably, *Hsd17b4* and *Hsd17b12a* were highly expressed in the testis and ovary, respectively, suggesting their involvement in the development of reproductive cells in these organs. Fluorescence in situ hybridization (FISH) further indicated specific expression sites for these genes, with *Hsd17b4* primarily expressed in germ stem cells and *Hsd17b12a* in oocytes. This comprehensive study provides foundational insights into the role of the Hsd17b gene family in gonadal development and steroidogenesis in *P. leopardus*, contributing to the broader understanding of fish reproductive biology and aquaculture breeding.

## 1. Introduction

Animal sex determination and differentiation are highly complex physiological processes. In most vertebrates, sex is regulated by conserved sex-determining genes [1]. However, in teleost fishes, methods of sex determination are diverse and plastic, with sex being determined by genetic factors, environmental factors [2], or a combination of both. Gonadal differentiation in fish begins in response to sex-determining signals, which induce the development of the gonads into either testes or ovaries [3]. Therefore, undifferentiated gonads have the bidirectional potential to differentiate into either testes or ovaries. Gonadal differentiation in teleosts takes various forms, including the direct differentiation and development of primordial gonads into testes or ovaries, as well as the presence of both functional male and female gonadal tissues in hermaphrodites [4]. As of now, the initial markers of testis and ovary differentiation in fish remain unclear.

The differentiation and developmental trajectory of fish gonads have always been a focal point and hot topic in the fields of fish genetics and developmental biology [5]. This research provides not only a theoretical basis for fish breeding and other technologies but also holds important implications for fish breeding and production enhancement. It was initially thought that the beginning of ovarian differentiation was marked by an increase in germ cells and the onset of meiosis to produce eggs [6]. However, this is not the case in the process of gonadal differentiation in zebrafish (*Danio rerio*). In larval *D. rerio*, there is a phenomenon of hermaphroditism, where ovary-like cells are initially formed and then decrease and disappear in gonads that develop into testes, followed by the development of testicular interstitial tissue [7]. Nowadays, the formation of the ovarian cavity is generally regarded as a histological sign of ovarian differentiation, and in most fish, the formation of the vas deferens signals the differentiation of the testis [8]. In addition to observing gonadal histology, with the advancement of molecular biology, many scholars also use sex-related genes to study the process of early gender differentiation, such as the female-related genes *cyp19a1a* [9] and *foxl2* [10], and the male-related genes *amh* [11], *dmrt1* [12], and *gsdf* [13].

Sex steroid hormones play a key role in sexual differentiation and the development of secondary sexual characteristics in fish [14]. As early as 1969, researchers discovered that estrogen promotes the formation of female characteristics during early gonadal development, while androgens promote the formation of male characteristics [15]. Although subsequent studies have shown that androgens are not the major internal regulators of differentiation in all male fish, the effects of estrogens and androgens on fish sex development are still significant [16]. 17β-Estradiol (E2) and testosterone (T) are two important sex hormones produced through the steroid synthesis pathway from a common starting material, cholesterol [17]. This process involves multiple steps, many of which are catalyzed by enzymes from the Hsd and Cyp families. Among these, 17β-Hsd, also known as Hsd17b, is a key enzyme in the synthesis of sex steroid hormones, acting downstream in the sex hormone synthesis signaling pathway [18]. It possesses both oxidative and reductive activities and is primarily responsible for the mutual conversion between estrone (E1), E2, androstenedione (A), and T [19]. To date, 15 types of HSD17Bs (Hydroxysteroid (17-beta) Dehydrogenases) have been identified in mammals, whereas in fish, the variety of these enzymes is relatively less. Currently, only 11 types of Hsd17b family genes have been reported in fish. In existing research on fish, Kazeto et al. successfully transfected the *Hsd17b1* gene from the Japanese eel into HEK 293 cells, where they observed its capability to convert E1 into E2 [20]. Furthermore, Rajakumar et al. discovered that during the gonadal development and maturation process in the walking catfish (*Clarias batrachus*), *Hsd17b1* exhibited high expression levels in the ovaries, suggesting a significant role for *Hsd17b1* in regulating sex hormone levels during gonadal development and gametogenesis [21]. Currently, the synthesis and metabolism pathway of sex steroid hormones in fish is a hot research topic. However, related research primarily focuses on the aromatase gene *cyp19a* [22] and some genes in the steroid hormone synthesis pathway, such as *StAR2* [23] and *cyp11a* [24]. Research on the Hsd17b family genes has been mainly concentrated in mammals, where their expression levels and genotypes are closely associated with various diseases [25]. However, there are relatively few reports on these genes in fish. Given the crucial role of the *Hsd17b* gene in regulating steroid synthesis and its varying functions across different species, it is particularly important to conduct in-depth research on the structure and function of the Hsd17b family genes in leopard coral grouper (*Plectropomus leopardus*).

The *P. leopardus*, belonging to the genus *Plectropomus* [26], is mainly found in tropical waters of the Indian and Pacific Oceans [27]. Due to its high nutritional value and excellent flavor, *P. leopardus* has become increasingly popular in the market. Research on *P. leopardus* has mainly been focused on aspects like growth [28], body color [29], nutrition [30], and pathology [31], with relatively few studies on gonad development and the sex determination mechanism of *P. leopardus*. This study intends to identify the Hsd17b family genes from the genome and transcriptome data of *P. leopardus* and conduct bioinformatics analysis on gene structure, chromosomal location, predicted protein characteristics, and related functions. Additionally, we constructed the expression profile analysis of the Hsd17b gene family across various tissues and different gonadal developmental stages in *P. leopardus*. This preliminary study on the role of these genes in the synthesis of sex steroid hormones is expected to provide foundational data for the synthesis and metabolism of fish sex steroid hormones and to lay the groundwork for elucidating the regulatory mechanism of fish gonad development.

## 2. Results

### 2.1. Amino Acid Sequence Alignment and Phylogenetic Analysis of the Hsd17b Gene Family

Through homology-based searches, 11 Hsd17b family genes were identified from the genome and transcriptome databases of the *P. leopardus*, including *Hsd17b1*, *-3*, *-4*, *-7*, *-8*, *-9*, *-10*, *-12a*, *-12b*, *-14*, and *-15*. Sequence alignment revealed that the average amino acid similarity among the *P. leopardus* Hsd17b family genes is only 12.96%, yet they still retain some conserved motifs, such as TGxxxGxG and S-Y-K (Appendix A).

Amino acid sequences of the Hsd17b family genes from 13 species were obtained from the NCBI database for phylogenetic tree construction (Figure 1A). The results showed that the Hsd17b family genes could be distinctly classified into six main branches and fourteen sub-branches. Branch I includes *Hsd17b8*, *-10*, *-14*, and *-4*; Branch II includes *Hsd17b1*, *-2*, *-6*, and *-9*; Branch III includes *Hsd17b11*, *-12a*, *-12b*, and *-13*; and Branches IV, V, and VI comprise *Hsd17b15*, *-7*, and *-5*, respectively.

Gene copy number analysis indicated that there are 15 members in the Hsd17b family, with *Hsd17b5* and *-6* being specific to humans (*Homo sapiens*) and mice (*Mus musculus*) (Figure 1B). Additionally, *Hsd17b11* and *-13* were not found in teleosts. *Hsd17b12* has only one copy in spotted gar (*Lepisosteus oculatus*) and common carp (*Cyprinus carpio*), whereas two subtypes are present in other teleosts.

### 2.2. Gene Structure Analysis

The analysis revealed significant variation in gene sequence lengths within the *P. leopardus* Hsd17b gene family (Figure 2A), with *Hsd17b15* being the shortest (3349 bp) and *Hsd17b4* the longest (41,166 bp). Eight conserved motifs were identified in these genes using MEME software (https://meme-suite.org/meme/tools/meme, accessed on 9 October 2023) (Figure 2B). For instance, *Hsd17b4*, *-8*, *-10*, and *-14*, located on the same branch, contain conserved motifs, such as TGxxxGxG, N, NNAG, S-Y-K, and PGxxxT. Similarly, *Hsd17b1* and *-9*, which are on the same branch, also possess these motifs. *Hsd17b3*, *-12a*, and *-12b* are on another branch, while *Hsd17b7* and -*15* are on their respective independent branches.

All members of the *P. leopardus* Hsd17b family belong to the SDR superfamily (Figure 2C). For example, *Hsd17b1*, *-9*, *-14*, and *-15* contain only the NADB_Rossmann superfamily domain; *Hsd17b3*, *-12a*, and *-12b* have the 17beta-HSD1_like_SDR_c domain; and *Hsd17b7*, *-8*, and *-10* each have a single domain, namely, 3KS_SDR_c, BKR_SDR_c, and HSD10-like_SDR_c, respectively. *Hsd17b4* is unique in having three conserved domains: hydroxyacyl-CoA-like_DH_SDR_c-like, PLN02864 superfamily, and SCP2.

### 2.3. Protein Structure and Biochemical Properties Prediction

The tertiary structures of proteins encoded by the *P. leopardus* Hsd17b gene family were predicted, as illustrated in Appendix A. All Hsd17b proteins in *P. leopardus* exhibit a helix–sheet–helix Rossmann structure, detailed in Appendix A.

The amino acid sequence predictions, presented in Appendix A, indicate that the smallest protein, *Hsd17b15*, comprises 228 amino acids, while the largest, *Hsd17b4*, contains 734 amino acids. The molecular weight range for the Hsd17b family proteins is from 25,391.26 Da to 79,034.15 Da, with theoretical isoelectric points (pI) varying from 5.8 to 9.7. Based on the GRAVY values, *Hsd17b4*, *-7*, and *-12a* are characterized as hydrophilic proteins due to their negative GRAVY values, whereas the others are identified as hydrophobic. The instability indexes of *Hsd17b3*, *-12b*, *-14*, and *-15*, all exceeding 40, suggest lower stability for these proteins.

### 2.4. Subcellular Localization

Subcellular localization predictions for the *P. leopardus* Hsd17b gene family indicate that *Hsd17b13*, *-7*, *-9*, *-12a*, *-12b*, and *-15* are located in the endoplasmic reticulum, *Hsd17b1* and *-14* in the cytoplasm, *Hsd17b8* and *-10* in the mitochondria, and *Hsd17b4* in the peroxisome. (Figure 3).

### 2.5. Chromosomal Distribution and Protein Interaction Prediction

The Hsd17b gene family in the *P. leopardus* is distributed across eight chromosomes (Figure 4A), namely, chr1 (*Hsd17b12b*), chr2 (*Hsd17b9*), chr4 (*Hsd17b7*), chr6 (*Hsd17b3* and *Hsd17b4*), chr7 (*Hsd17b10*), chr11 (*Hsd17b12a*), chr17 (*Hsd17b1* and *Hsd17b14*), and chr 18 (*Hsd17b8*).

Predictions of protein–protein interactions within the *P. leopardus* Hsd17b family reveal significant interactions (Figure 4B). For example, the homologs of *Hsd17b12a* and *-8* are co-expressed with *Hsd17b1*, *-7*, and *-12b*, while the homologs of *Hsd17b12a* and *-4* are co-expressed with *Hsd17b7* and *-8*. Cluster analysis shows that *Hsd17b1*, *-3*, *-7*, *-8*, *-12a*, and *-12b* group together.

### 2.6. Expression Levels of Hsd17b Family Genes in Male and Female Gonads at Different Stages

In *P. leopardus*, the Hsd17b gene family exhibits significant differences in expression during various stages of gonadal development (Figure 5A). These genes can be broadly classified into three categories: the first category includes *Hsd17b3*, *-4*, *-9*, *-12b*, and *-14*, which are generally more highly expressed in the testes than in the ovaries; the second category includes *Hsd17b8*, -*10*, and-*12a*, with higher expression in the ovaries than in the testes; and the third category includes *Hsd17b1*, *-7*, and *-15*, with no significant difference in expression between the male and female gonads. Additionally, it was observed that from 120 dph to 3YT, the expression level of *Hsd17b4* gradually increases in the testes, significantly exceeding that in the ovaries. Conversely, the expression level of *Hsd17b12a* gradually increases in the ovaries, significantly exceeding that in the testes. *Hsd17b1* and *-7* are expressed in all stages except the BiT stage, while *Hsd17b15* is expressed at a low level in all stages.

### 2.7. Expression Levels of Hsd17b Family Genes in Different Tissues

Expression levels of the *P. leopardus* Hsd17b family genes vary across different tissues (Figure 5B), mainly concentrated in the liver, heart, gills, kidneys, and gonads. *Hsd17b1* exhibits elevated expression in the spleen. *Hsd17b12a* shows high expression in the gills, brain, and ovaries. *Hsd17b7* is predominantly expressed in the gills and ovaries. The liver expresses the highest levels of *Hsd17b14*. *Hsd17b4* and *Hsd17b12b* share a similar expression profile, being highly expressed in the kidney, liver, brain, gills, testes, and intestine. Similarly, *Hsd17b8* and *Hsd17b10* demonstrate comparable patterns, with elevated expression in the gonads, kidneys, and heart. Both *Hsd17b9* and *Hsd17b15* display lower expression across all examined tissues.

### 2.8. HE Staining Observation and Specific Expression of Hsd17b4 and -12a in Gonads

Histological sections of *P. leopardus* gonads at different developmental stages show that at 120 days post-hatch (dph), the gonads are undifferentiated, consisting predominantly of germ stem cells (Gscs) (Figure 6A). During the bisexual phase (Bi), the bisexual testis (BiT) phase is mainly composed of the primary spermatocyte phase (Psc), spermatocytes (Sc), and a small number of spermatids (St). In the bisexual ovary (BiO) phase, the number and diameter of oocytes increased, mainly stage I and II oocytes, and the number of marginal male germ cells decreased. At 3 years in males (3YT), the gonads contain all stages of male germ cells but are dominated by Sc and mature St, whereas in females of the same age (3YO), the gonads predominantly contain stage III and IV oocytes.

The results show that *Vasa* has fluorescent signals in Gscs, oocytes, Psc, Sc, and St, but there is no fluorescent signal in somatic cells, while *Hsd17b4* mainly has fluorescent signals in Gscs (Figure 6(Bb,Bc)). Using BIO and DIG-labeled probes for *Vasa* and *Hsd17b12a* RNA, it was observed (Figure 6(Bd,Be)) that *Hsd17b12a* primarily exhibited fluorescence signals in oocytes, predominantly in stages I and II, with vasa showing signals in both male and female gametes. The detection results of RNA probes labeled *Hsd17b4* and *Hsd17b12a* using BIO and DIG, respectively (Figure 6(Bf,Bg)), show that the *Hsd17b4* of *P. leopardus* has a stronger fluorescence signal detected in BiT than BiO. *Hsd17b12a* has a stronger fluorescence signal detected in BiO than BiT.

## 3. Discussion

The gender of fish exhibits significant plasticity, being influenced not only by genetic factors but also potentially by certain intrinsic or extrinsic factors, particularly sex steroids [22]. Hsd17b enzymes play a crucial role in the biosynthesis of sex steroids, acting downstream in the hormone synthesis pathway and involved in the interconversion of E1, E2, A, and T [19]. To date, 15 Hsd17b family genes have been reported in *H. sapiens* and *M. musculus* [32], while fewer have been identified in fish, with eleven reported in olive flounder (*Paralichthys olivaceus*) and only nine in other fish species [33]. In the current research, 11 Hsd17b family genes were identified from the genome and transcriptome data of *P. leopardus*, with *Hsd17b9* and *-15* previously reported only in *P. olivaceus*. Additionally, it was found that *Hsd17b12* has two homologs in fish, *Hsd17b12a* and *-12b*, located on chromosomes 11 and 1, respectively, indicating they are two independent genes, not different transcripts of the same gene. Although Hsd17b family genes, except for *Hsd17b5*, which belongs to the AKR family, are part of the SDR superfamily, their amino acid similarity is relatively low, typically only 15% to 20% in vertebrates [25]. The amino acid sequence alignment of the Hsd17b family genes in the *P. leopardus* revealed only a 12.96% similarity. Despite this, these sequences still possess important conserved motifs, such as the coenzyme binding region (TGxxxGxG), active site (S-Y-K), reaction direction (PGxxxT), and the N-N-A-G motif, crucial for coenzyme binding and catalytic reaction direction [34].

Based on this, we further clarified the relationships within the Hsd17b gene family using phylogenetic analysis. The phylogenetic tree indicates that even the same gene members exhibit differences in kinship across different species. The amino acid sequences of the Hsd17b family genes in the *P. leopardus* show the closest evolutionary relationships with those in teleosts, such as grouper (*Epinephelus lanceolatus*), flounder (*Paralichthys dentatus*), Nile tilapia (*Oreochromis niloticus*), and Japanese Killifish (*Oryzias latipes*), while being more distantly related to tetrapods, like *H. sapiens* and *M. musculus*. This is consistent with the findings of London and Clayton [35] in their phylogenetic studies of mammals, birds, and fish. *Hsd17b3* and *-12*, as well as *Hsd17b1*, *-2*, *-6*, and *-9*, group together, a pattern also confirmed in the research by Mindnich et al. [36]. In human studies, *Hsd17b3* and *-12* share similar structures, likely resulting from gene duplication, and are located on different chromosomes. Our results largely align with those found in flounder, yet differ in that *Hsd17b4* clusters with *-8*, whereas in the study by London and Clayton [35], *Hsd17b8* clusters with *-14*, with a low confidence value at this branch. Initially, *Hsd17b8* was classified within the Hsd17b family primarily due to its high similarity to *Hsd17b4* [37]. Furthermore, the inclusion of *Hsd17b15* in the phylogenetic tree construction might also contribute to the observed differences in branching.

Although the protein of Hsd17b family genes have similar structures, they exhibit significant differences in expression specificity, subcellular localization, participation in biological processes, and catalytic activity across different tissues [38]. These variations play a critical role in regulating intracellular levels of estrogens and androgens in peripheral cells [39]. The enzymes encoded by the Hsd17b family genes can be categorized into two classes based on their catalytic properties: oxidases and reductases [40]. The oxidase enzymes include *Hsd17b2*, *-4*, *-8*, *-9*, *-10*, and *-14*, while the reductase enzymes comprise *Hsd17b1*, *-3*, *-5*, *-7*, *-11*, and *-12*. These enzymes are primarily involved in the biosynthesis and metabolism of E2 and T. In mammals, *Hsd17b2*, *-4*, *-8*, and *-14* are key in converting E2 to E1 [41], whereas *Hsd17b3* and *-5* play important roles in androgen synthesis, with *Hsd17b3* being closely related to the development of male reproductive characteristics. Research on Hsd17b family genes has mainly been focused on mammals in both biological and pathological contexts, as imbalances in sex steroid hormones are closely linked to the pathogenesis of various human diseases, such as fatty liver [42], prostatitis, breast cancer [43], and Alzheimer’s disease [44]. Additionally, these genes participate in diverse biological pathways, indicated by the differences in target sites among various genotypes. Similar patterns of differential expression are observed in fish, with notable variances between genotypes in different tissues and between sexes. For instance, the cDNA sequence of *Hsd17b1* has been cloned in species like *D. rerio* [36], freshwater eel (*Anguilla japonica*) [20], and flounder, with studies showing its activity in converting E1 to E2. Current research validates higher expression levels of Hsd17b1 in the ovaries compared to the testes. In *O. niloticus*, *Hsd17b3* was found to be exclusively expressed in the testes and only during the onset of gametogenesis [45]. Additionally, *Hsd17b8* was found to facilitate the mutual conversion of E2 and E1 in *O. niloticus*. In the present study, *Hsd17b3* was almost not expressed in the ovary of *P. leopardus*, and the expression of *Hsd17b8* was higher in the ovary than in the testis. The results of tissue-specific expression suggested that *Hsd17b4*, *-10*, *-12b*, and *-14* in the *P. leopardus* are predominantly expressed in the liver, a crucial site for various metabolic activities, including the metabolism of sex steroid hormones [46]. Additionally, the heart, gills, kidneys, and brain are also key expression sites for the Hsd17b gene family, suggesting potential roles in the synthesis or metabolism of steroids [47] and neurotransmitters, which warrant further experimental validation [48]. Overall, the Hsd17b gene family is vital in maintaining hormonal balance within organisms.

Based on the observation of the expression patterns of the Hsd17b gene family in the gonads of *P. leopardus* at different stages of gonad development, they can be divided into three categories. The first category is generally expressed in the testes higher than in the ovaries, including *Hsd17b3*, *-4*, *-12b*, and *-14*. The second category is expressed in ovaries higher than testes, including *Hsd17b8*, *-10*, and *-12a*. The third category has no significant expression difference in male and female glands, including *Hsd17b1*, *-7*, and *-15*. In the study of *P. olivaceus*, Zhou et al. [33] also observed that the expression patterns of the Hsd17b gene family during different gonadal development stages can be broadly categorized into three main types, similar to our findings in *P. leopardus*. Furthermore, by combining gene family analysis with transcriptomic data of *P. leopardus*, we identified *Hsd17b4* and *Hsd17b12a* as genes differentially expressed between sexes. *Hsd17b4* was found to be predominantly expressed in males, potentially associated with the development of male gametes, while *Hsd17b12a* showed higher expression in females, likely related to female gamete development. Through fluorescence in situ hybridization experiments, we validated the localization of these genes. The results indicated that *Hsd17b4* is primarily expressed in germ stem cells, whereas *Hsd17b12a* is mainly expressed in oocytes. These findings suggest that *Hsd17b4* and *Hsd17b12a*, as key enzymes in steroid hormone synthesis, play a critical role in the sexual differentiation of fish.

Studies on *Hsd17b4* are relatively scarce, with its functional involvement in sex steroid metabolism, fatty acid β-oxidation, sterol transport, and bile acid biosynthesis [49,50]. *Hsd17b4* is predominantly expressed in the liver, kidney, brain, and gonadal organs. The overexpression of *Hsd17b4* in human studies has been linked to prostate cancer, suggesting its potential as a biomarker for androgen metabolism [51]. Furthermore, *Hsd17b4* regulates steroid concentration through catalytic oxidation or reduction at the C17 position. Pyun et al. [52] reported that the interaction between *Hsd17b4* and thyroglobulin (TG) is associated with premature ovarian failure. *Hsd17b4*, with its unique three-domain structure and localization in cellular peroxisomes, is involved in fatty acid metabolism, and mutations in its gene may lead to Zellweger syndrome [53]. There are few reports on *Hsd17b4* in fish research. Among the reports on Japanese medaka (*Oryzias latipes*), *Hsd17b4* may play an important role in the gonadal differentiation process [54]. In the current research, the expression of *Hsd17b4* in *P. leopardus* varied with sex development, being significantly higher in the testes than in the ovaries, indicating a potential role in promoting testicular development. *Hsd17b12* is an endoplasmic reticulum-bound ketoacyl reductase, catalyzing the conversion between E1 and E2. In mammals, *Hsd17b12*, along with *-1* and *-7*, has been reported to convert E1 to E2, and in rodents, it is also involved in converting A to T [55]. Despite controversies over its role in converting E1 to E2, this function has been confirmed in cell transfection and RNAi interference experiments. Bellemare et al.’s [56] mouse knockout studies showed that hybrids (HSD17B12+/−) exhibited significantly reduced sex steroid hormone levels. This was also confirmed by Laplante et al. [57] in studies of human breast cancer cell lines, whose inhibitors had a clear inhibitory effect on the conversion of E1 to E2. While the function of *Hsd17b12* in converting E1 to E2 has been extensively studied in mammals, reports in fish are scarce. Mindnich et al.’s [58] research on *D. rerio* suggested that both *Hsd17b12a* and *-12b* are involved in converting E1 to E2. Through this exploration, the domains of *Hsd17b12a* and *-12b* in *P. leopardus* were identified as 17beta-HSD1_like_SDR_c, indicating a tendency for E1 reduction. Human studies have confirmed the high estrogen specificity of *Hsd17b1* with low catalytic capacity for androgen substrates. GO annotation of fish genes based on the human gene database revealed that both *Hsd17b1* and *-12* are involved in the biological process of estrogen biosynthesis [33]. This study speculates that *Hsd17b12* in *P. leopardus* may have similar catalytic functions to *Hsd17b1*. Further research is needed on the function of *Hsd17b12* in fish and its role in sex hormone metabolism pathways.

## 4. Materials and Methods

### 4.1. Experimental Fish and Sample Collection

The *P. leopardus* used in the experiment were all purchased from Hainan Chenhai Co., Ltd. (Hainan, China). Based on the original experiments of the research group, gonadal samples were collected at three developmental stages: the gonadal undifferentiated stage (120 days post-hatch [dph]), the gender facultative stage (15 months old), and the sexually mature stage (3 years old). Prior to sampling, the fish were anesthetized using eugenol at a concentration of 30 mg/L, and 15 individuals were selected at each stage for gonad removal. Half of each sample was fixed in 4% paraformaldehyde for subsequent Hematoxylin and Eosin (HE) staining and in situ hybridization. The other half was stored in liquid nitrogen for subsequent RNA extraction. 

### 4.2. Identification and Phylogenetic Analysis of Hsd17b Family Members

To identify the Hsd17b gene family members of *P. leopardus*, joint searches were conducted on the NCBI (http://www.ncbi.nlm.nih.gov, accessed on 9 October 2023) and UniProt (https://www.uniprot.org/, accessed on 9 October 2023) websites. The amino acid sequences of the Hsd17b gene family are presented in Appendix A. The gene sequences of Hsd17b family members were screened through local BLAST (e-value = 1 × 10^−5^) against the laboratory’s existing *P. leopardus* genome (NCBI accession number GCA_008729295.2) and transcriptome database [59]. Subsequently, MEGA 11.0 software was used to construct a phylogenetic tree with a bootstrap value of 1000 [60]. The results of the phylogenetic tree construction were visualized and refined using the iTOL website (https://itol.embl.de/, accessed on 9 October 2023). The genome annotation file was utilized to locate the SSR locus on the chromosome of *P. leopardus*, and the results were analyzed using TBtools [61].

### 4.3. Physicochemical Properties Analysis of Hsd17b Family Members

The amino acid sequences of the Hsd17b family in the *P. leopardus* were aligned, and the average amino acid identity values were calculated using DNAMAN 6.0 (http://www.lynnon.com). The prediction of conserved domains and motifs in the genes of *P. leopardus* was carried out using CDD NCBI (https://www.ncbi.nlm.nih.gov/cdd, accessed on 11 October 2023) and MEME (http://meme-suite.org/index.html, accessed on 11 October 2023), respectively. The gene structure and protein structure of the *P. leopardus* Hsd17b family genes were obtained using GSDS 2.0 (http://gsds.gao-lab.org/, accessed on 11 October 2023) and SWISS-MODEL (https://swissmodel.expasy.org/, accessed on 12 October 2023). Subcellular localization of the proteins was predicted using the method available at http://www.csbio.sjtu.edu.cn/bioinf/Cell-PLoc-2/, accessed on 12 October 2023. The biochemical properties of the proteins were predicted using ExPASy (http://web.expasy.org/protparam/, accessed on 13 October 2023), and a protein interaction network map was constructed using STRING 11.0 (https://string-db.org/, accessed on 13 October 2023). Cluster analysis was conducted based on the k-means method with a clustering coefficient of 6 [33].

### 4.4. Transcriptome-Based Expression Analysis of Hsd17b

The expression profiles of Hsd17b in healthy *P. leopardus* were analyzed using transcriptome datasets. A total of 33 datasets, which cover 11 different tissues (heart, liver, spleen, kidney, brain, gill, muscle, intestine, skin, testis, and ovary), were utilized to calculate the Transcripts Per Million (TPM) values for Hsd17b (NCBI accession number PRJNA1012450). In addition, the expression pattern of Hsd17b in individual gonads at different developmental stages was examined using RNA-seq datasets from our laboratory. Using GraphPad Prism software (https://www.graphpad.com/features, accessed on 16 October 2023), heatmaps were generated with logarithmic transformation (Log10) and row-wise normalization to clearly illustrate the expression differences of the Hsd17b gene family across various tissues.

For further validation, we extracted RNA from gonadal tissues using the TRIzol method, ensuring high quality (RIN > 7). For mRNA sequencing, we employed bead-based mRNA isolation and sequencing on the Illumina platform. Differential gene expression was analyzed with DESeq, focusing on genes with significant changes in expression (log2 fold change greater than 1, *p*-value less than 0.05). To validate these findings, selected differentially expressed genes (DEGs) were further examined using qPCR on the Applied Biosystems 6300 RT-PCR system [62]. During qPCR analysis, the β2-microglobulin (b2m) gene was used as an internal control, and the relative expression levels were quantified using the 2^−ΔΔCt^ method [63], ensuring the reliability of the transcriptomic data.

### 4.5. Histological Observation

The detailed experimental procedures have been summarized in our previous study [64]. Briefly, the gonad samples were fixed in 4% paraformaldehyde at 4 °C overnight, dehydrated with ethanol, and then embedded in paraffin. Tissue blocks were cut into 5 μm thick sections using a microtome and stained with the Hematoxylin and Eosin Staining Kit (Beyotime, Haimen, China).

### 4.6. Fluorescence In Situ Hybridization (FISH)

RNA probe synthesis was performed using the DIG RNA Labelling Kit (Roche, Mannheim, Germany) following the manufacturer’s protocol. Primers used for probe template amplification are listed in Appendix A. The detailed procedure was conducted according to the protocol from our laboratory [65]. Gonad tissues of *P. leopardus* during bisexual testis (BiT) and bisexual ovary (BiO) stages were selected for the localization of *Hsd17b4*, *Hsd17b12a*, and *Vasa* RNA via fluorescence in FISH. All DNA was stained blue with diamidino-phenyl-indole (DAPI). Biotin (BIO) and digoxigenin (DIG) were used to label *Hsd17b4* and *Vasa* RNA probes, respectively, for distribution in the Bi-phase gonads of *P. leopardus*. FISH results were viewed and photographed using an Olympus FV3000 confocal microscope (Olympus, Tokyo, Japan).

### 4.7. Statistical Analysis

Statistical analysis was performed using GraphPad Prism 9 (GraphPad Software, Inc., San Diego, CA, USA). Before applying parametric statistical tests, we checked the normality and homoscedasticity of the data to ensure that the underlying assumptions were met. All experimental data were presented as the mean ± SEM with an *n*-value of at least three. Differences between groups were calculated using one-way ANOVA followed by an independent sample *t*-test or a Games–Howell test (SPSS 16.0, New York, NY, USA). A *p*-value < 0.05 was considered to indicate a significant difference between groups.

## 5. Conclusions

This research identifies and characterizes the Hsd17b gene family in *P. leopardus*, belonging to the SDR superfamily, with distinct expression in tissues, like the liver, heart, and gonads, suggesting their role in the synthesis or metabolism of sex steroids and neurotransmitters. This family of genes exhibits diverse expression patterns during different gonadal development stages, with *Hsd17b4* and *Hsd17b12a* showing high expression in the testis and ovary, respectively, suggesting their roles in the development of reproductive cells in these organs. Further, the FISH confirmed their specific expression sites, highlighting the Hsd17b gene family’s significant roles in gonadal development. This study advances our understanding of fish sex steroid synthesis and the regulatory mechanisms of gonadal development.

## Figures and Tables

**Figure 1 ijms-25-02180-f001:**
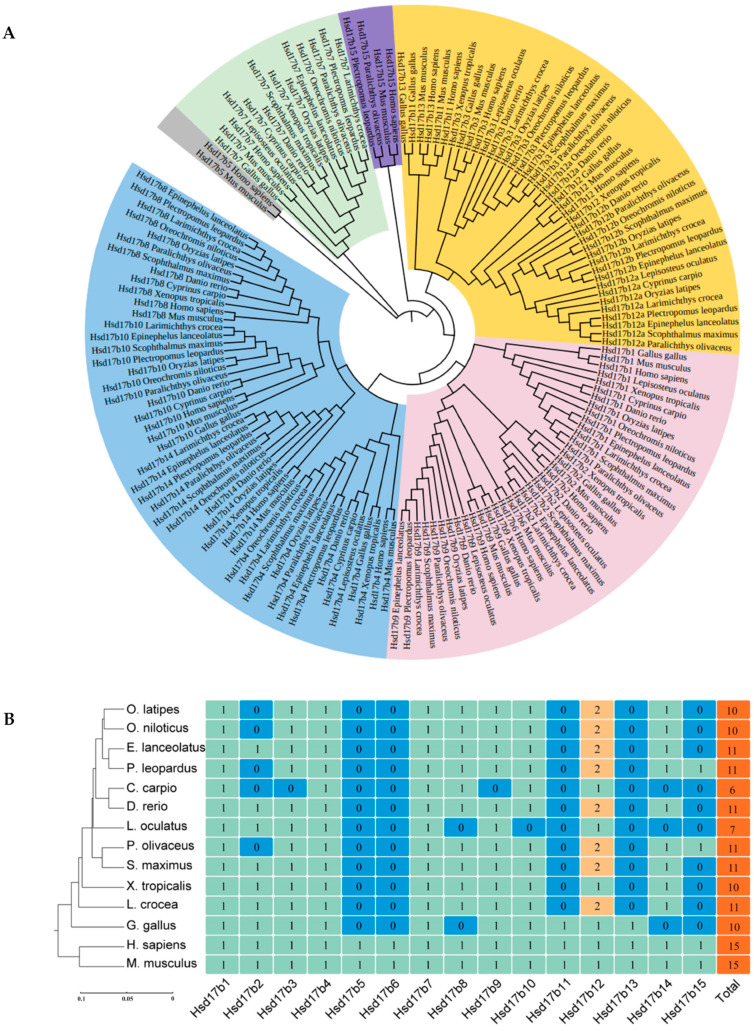
Phylogenetic tree (**A**) and gene copy number (**B**) analysis of the Hsd17b gene family in *P. leopardus* and other species. Note: The phylogenetic tree was constructed by MEGA 11.0 using the neighbor joining method, and the confidence values higher than 70 were indicated on the tree.

**Figure 2 ijms-25-02180-f002:**
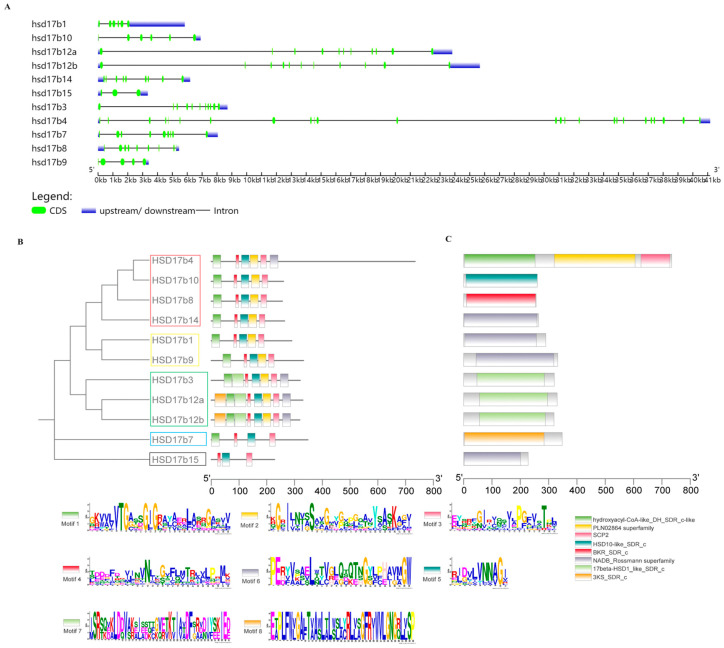
Gene structure (**A**), motif analysis (**B**), and conserved domains (**C**) of the Hsd17b gene family in *P. leopardus*. Note: In (**B**), red boxes represent Hsd17b genes belonging to Clade I; yellow boxes represent *Hsd17b* genes belonging to Clade II; green boxes represent *Hsd17b* genes belonging to Clade III; black boxes represent *Hsd17b* genes belonging to Clade IV; blue boxes represent *Hsd17b* genes belonging to Clade V. Boxes with different colors indicated different motifs and domains.

**Figure 3 ijms-25-02180-f003:**
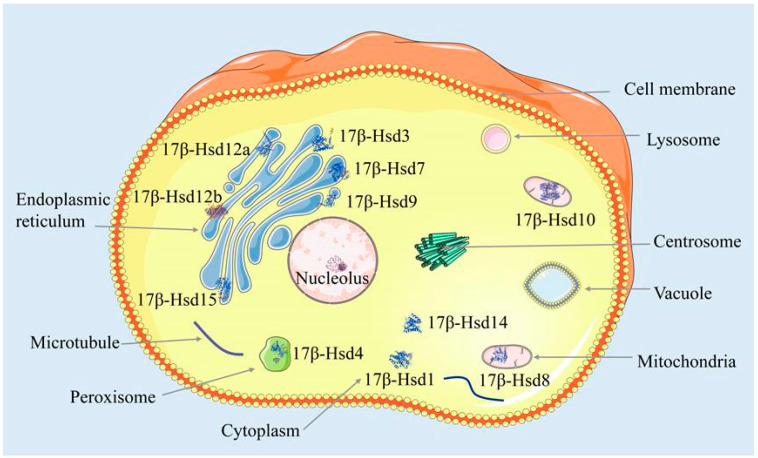
Subcellular localization of the Hsd17b gene family in *P. leopardus*.

**Figure 4 ijms-25-02180-f004:**
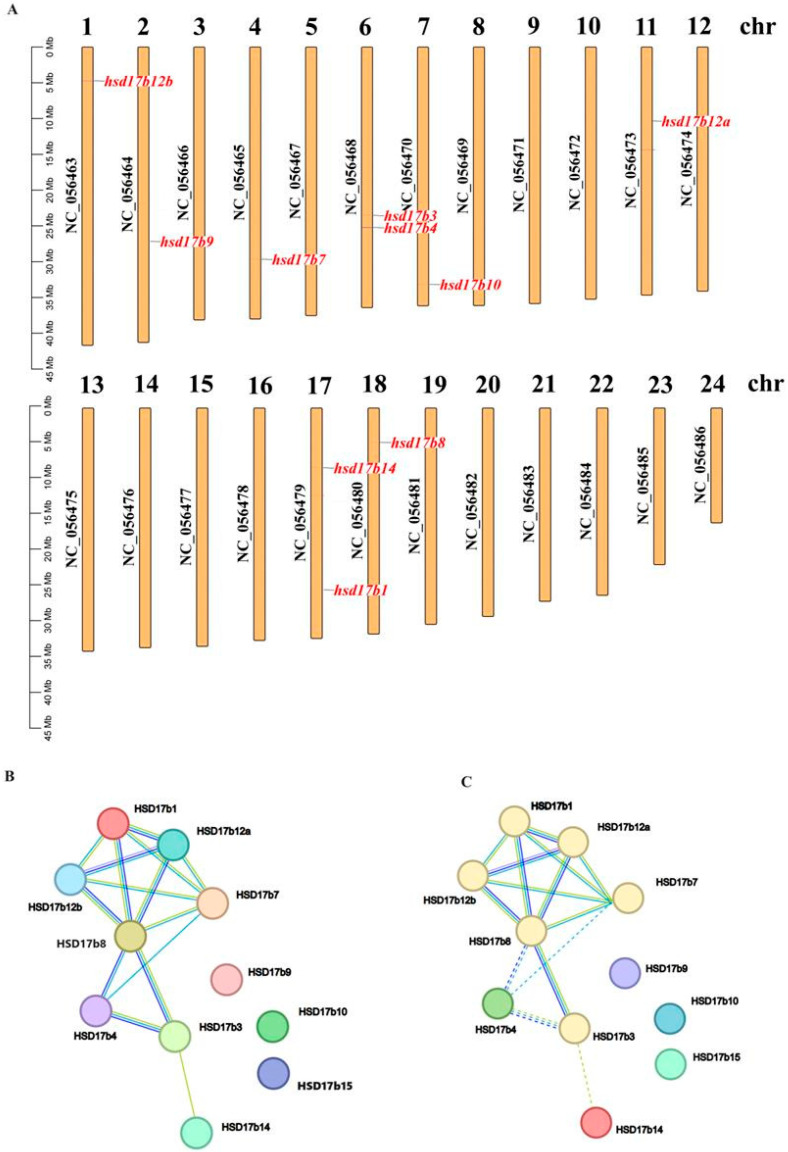
Chromosomal distribution (**A**) and protein–protein interaction analysis (**B**,**C**) of the Hsd17b gene family in *P. leopardus*. Note: (**B**) original network, (**C**) network clustered to 6 clusters.

**Figure 5 ijms-25-02180-f005:**
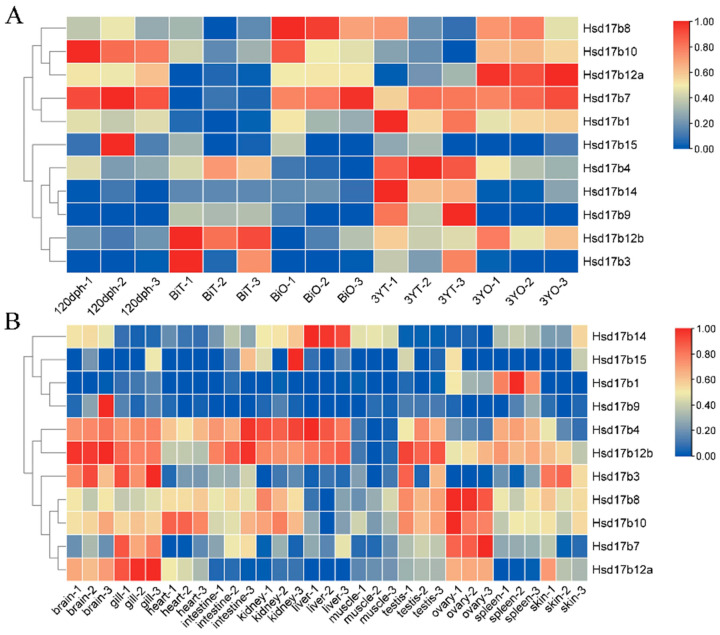
Expression profiles of the Hsd17b gene family in *P. leopardus* across different developmental stages (**A**) and all tissues (**B**). Note: 120 dph: 120 days post-hatch, the gonadal undifferentiated stage; BiT: 15-month-old testis in the bisexual phase; BiO: 15-month-old ovary in the bisexual phase; 3YT: 3-year-old testis; 3YO: 3-year-old ovary. Red, high expression. The color scale indicated the TPM values. The colors from blue to red represent the lower TPM values to higher TPM values.

**Figure 6 ijms-25-02180-f006:**
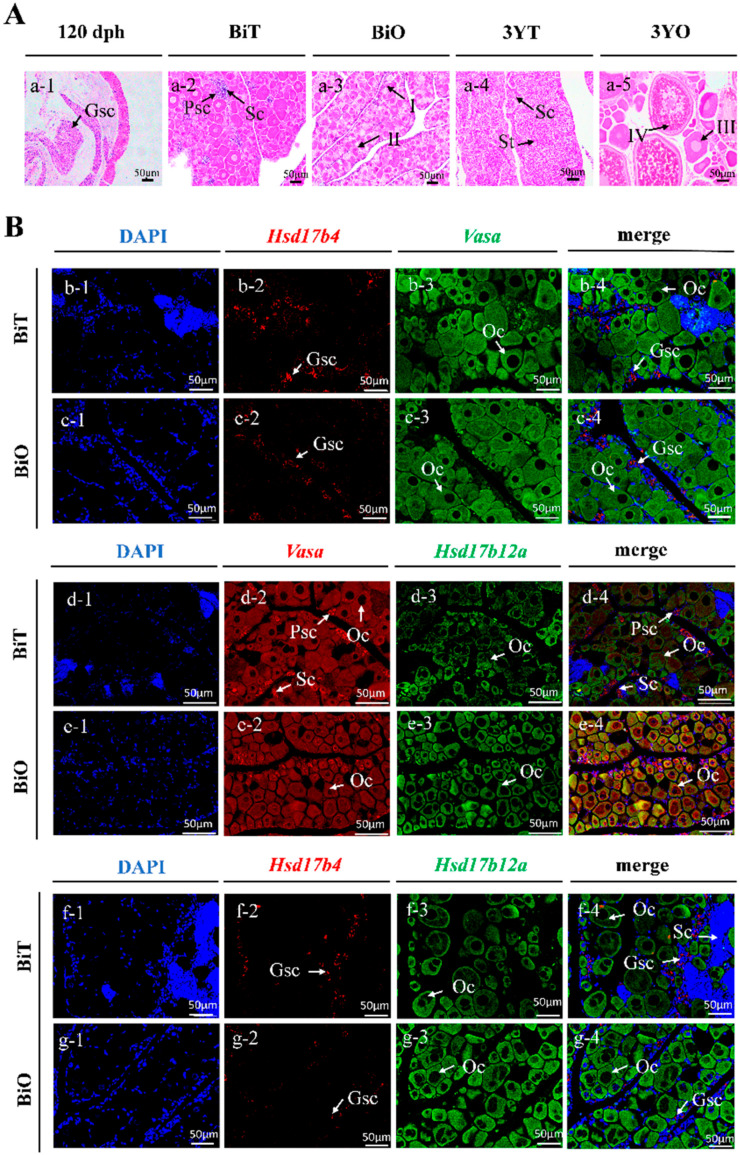
Localization of *Hsd17b4* and *-12a* RNA by fluorescence in situ hybridization (FISH) and histology. Note: (**Aa-1**–**Aa-5**) includes stages of 120 days post-hatch (120 dph), bisexual testis (BiT) phase, bisexual ovary (BiO) phase, 3-year-old males (3YT), and 3-year-old females (3YO) with HE staining of gonads (20× magnification). Gonad tissue from the *P. leopardus* in the BiT and BiO stages was used for localizing *Hsd17b4*, *Hsd17b12a*, and *Vasa* RNA via FISH experiments in (**B**) (40× magnification). Stain all DNA blue (**b-1,c-1,d-1,e-1,f-1,g-1**) with diamidino-phenyl-indole (DAPI). Digoxigenin (DIG) stains green (*Vasa*: **b-3,c-3**; *Hsd17b12a*: **d-3,e-3,f-3,g-3**) and biotin (BIO) stains red (*Hsd17b4*: **b-2,c-2,f-2,g-2**; *Vasa*: **d-2,e-2**). Merge is a merged image (**b-4,c-4,d-4,e-4,f-4,g-4**). Staining was analyzed by confocal microscopy. Gscs: germ stem cells; I, early previtellogenic phase; II, late previtellogenic phase; III, early vitellogenic phase; IV, fully vitellogenic phase; Oc: oocyte; Psc: primary spermatocyte phase; Sc, spermatocytes; St, spermatids. The scale bar is located at the bottom right corner of the figure, representing 50 μm.

## Data Availability

The data has not been published yet and is delayed to be released. If you need original data, please contact the corresponding author.

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
