# Peer review of "Theoretical Analysis and Expression Profiling of 17β-Hydroxysteroid Dehydrogenase Genes in Gonadal Development and Steroidogenesis of Leopard Coral Grouper (Plectropomus leopardus)"

_ijms, 2024, doi:10.3390/ijms25042180_

Round 1

Reviewer 1 Report

Comments and Suggestions for Authors

This manuscript written by Liu et al. explores the role of the 17β-Hydroxysteroid Dehydrogenase (Hsd17b) gene family in the gonadal development and steroidogenesis of the leopard coral grouper, providing key insights into fish reproductive biology and aquaculture breeding, especially for the application in fish. This paper can evidently attract the interest of readers. However, there are a few minor issues that need improvement.

Comments:

1.     17β-Hsd is a key enzyme in the synthesis of sex steroid hormones, acting downstream in the sex hormone synthesis signaling pathway. However, in the introduction, there is no descriptions of existing studies of 17β-Hsd in fish.

2.     On line 200, the species name P. leopardus needs to be italicized.

3.     In Figure 6.c-2, the orientation of the in situ hybridization image needs to be adjusted.

4.     L247 “only 9 in other fish species” – which species?

5.     L263 “grouper” please give the Latin name and the specific species name.

6.     L276 "Although the Hsd17b family genes have similar protein structures" is suggested to be changed to: "Although the protein of Hsd17b family genes have similar structures."

7.     Please provide the concentration of eugenol on line 365.

8.     How to achieve subcellular localization of the Hsd17b gene family?

9.     L403 Please give the version number of GraphPad.

10.  The conclusion is too verbose and could be made more concise.

11.  The Latin names of some species in the references are not italicized. Please check.

Author Response

We gratefully thank the editor and all reviewers for their time spend making their constructive remarks and valuable suggestions, which has significantly improved the quality of the manuscript and has enabled us to improve the manuscript. Each suggested revision and comment brought forward by the reviewers was accurately incorporated and considered. Below the comments of the reviewers are responses point by point, and the revisions are indicated. Modifications are marked with red color.

This manuscript written by Liu et al. explores the role of the 17β-Hydroxysteroid Dehydrogenase (Hsd17b) gene family in the gonadal development and steroidogenesis of the leopard coral grouper, providing key insights into fish reproductive biology and aquaculture breeding, especially for the application in fish. This paper can evidently attract the interest of readers. However, there are a few minor issues that need improvement.

Comments:

  1. 17β-Hsd is a key enzyme in the synthesis of sex steroid hormones, acting downstream in the sex hormone synthesis signaling pathway. However, in the introduction, there is no descriptions of existing studies of 17β-Hsd in fish.

Response: Thank you very much for your valuable comments. We added this section. as follows: In fish studies, Kazeto et al. (2000a) successfully transfected the Hsd17b1 gene from the Japanese eel into HEK 293 cells, where they observed its capability to convert E1 into E2. Furthermore, Rajakumar et al. (2014) discovered that during the gonadal development and maturation process in the walking catfish (C. batrachus), Hsd17b1 exhibited high expression levels in the ovaries, suggesting a significant role for Hsd17b1 in regulating sex hormone levels during gonadal development and gametogenesis. line 81-89.

  1. On line 200, the species name P. leopardus needs to be italicized.

Response: Thanks for your suggestion, we have corrected this error.

  1. In Figure 6.c-2, the orientation of the in situ hybridization image needs to be adjusted.

Response: Thanks for your suggestion, we have corrected this error.

  1. L247 “only 9 in other fish species” – which species?

Response: In the context of Hsd17b gene family, comprehensive documentation exists only for humans and laboratory mice, comprising a total of 15 genes. Among fish species, research efforts have predominantly focused on the Olive flounder (Paralichthys olivaceus), where 11 family genes have been reported. For other fish species, a lesser number of family genes, specifically 9, have been documented. In this study, we chose to compare Hsd17b gene family members in various fish species, including D. rerio, S. maximus, L. oculatus, O. latipes, L. crocea, O. niloticus, E. lanceolatus, and C. carpio. Given the numerous Latin names involved, we refrain from listing them individually in the discussion for conciseness, as per the language requirements of scientific publications.

  1. L263 “grouper” please give the Latin name and the specific species name.

Response: Thanks for your suggestion, we have corrected this error. The name of the grouper is Epinephelus lanceolatus.

  1. L276 "Although the Hsd17b family genes have similar protein structures" is suggested to be changed to: "Although the protein of Hsd17b family genes have similar structures."

Response: Thanks for your suggestion, we have corrected this error.

  1. Please provide the concentration of eugenol on line 365.

Response: Thank you for your review and valuable suggestions on our paper. We have modified this part. Modify as follows: Prior to sampling, the fish were anesthetized using eugenol at a concentration of 30 mg/L, and 15 individuals were selected at each stage for gonad removal. Line:375-377

  1. How to achieve subcellular localization of the Hsd17b gene family?

Response: Thank you for your review and valuable suggestions on our paper. We understand your concerns very much. Subcellular localization of the proteins was predicted using the method available at http://www.csbio.sjtu.edu.cn/bioinf/Cell-PLoc-2/. Line:399-401

  1. L403 Please give the version number of GraphPad.

Response: Thank you for your review and valuable suggestions on our paper. We have modified this part. Modify as follows: Statistical analysis was performed using GraphPad Prism 9 (GraphPad Software, Inc., California, USA). Line:444

  1. The conclusion is too verbose and could be made more concise.

Response: Thank you for your review and valuable suggestions on our paper. We have modified this part. Modify as follows:

This research identifies and characterizes the Hsd17b gene family in P. leopardus, belonging to the SDR superfamily, with distinct expression in tissues like liver, heart, and gonads, suggesting their role in the synthesis or metabolism of sex steroids and neurotransmitters. This family genes exhibit diverse expression patterns during different gonadal development stages, with Hsd17b4 and Hsd17b12a showing high expression in the testis and ovary, respectively, suggesting their roles in the development of reproductive cells in these organs. Further, the FISH confirmed their specific expression sites, highlighting the Hsd17b gene family's significant roles in gonadal development. This study advances our understanding of fish sex steroid synthesis and the regulatory mechanisms of gonadal development. Line:452-461

  1. The Latin names of some species in the references are not italicized. Please check.

Response: Thanks for your suggestion, we have corrected this error.

Reviewer 2 Report

Comments and Suggestions for Authors

Dear authors,

This work tries to summarize the expression of Hsd17b gene family in P. leopardus tissues, and theoretical analysis of function and phylogeny. The manuscript is well written and structured, the introduction provides sufficient background, and the cited references are relevant to the research. However, some changes are necessary before to publish. First, the title should be modified to explain that the function and phylogenetic analysis is a theoretical analysis, since the authors have only carried out an expression analysis. On the other hand, the authors must improve the material and methods section, since they have not explained how the gene expression analysis was carried out. They only indicate that the data have been obtained from previous unpublished studies from their laboratory, but they must explain how they obtained the samples, how they obtained the RNA, what method they used to carry out the gene expression analysis, and what analysis methods. statistics have used. In addition, and more specifically, other changes are necessary such as:

-          Lines 64, 81 and 92: eliminate “etc”. If other genes exist, the author should enumerate it and cited references.

-          Line 84: the authors should refer to the gene family?

-          Line 369: specify the method and quantities used to anesthetize the animals.

-          Line 372: “in situ” in italics.

-          Section “Statistical analysis”: the authors should check the normality and homoscedasticity of data before to use parametric statistical test.

Author Response

We gratefully thank the editor and all reviewers for their time spend making their constructive remarks and valuable suggestions, which has significantly improved the quality of the manuscript and has enabled us to improve the manuscript. Each suggested revision and comment brought forward by the reviewers was accurately incorporated and considered. Below the comments of the reviewers are responses point by point, and the revisions are indicated. Modifications are marked with red color.

This work tries to summarize the expression of Hsd17b gene family in P. leopardus tissues, and theoretical analysis of function and phylogeny. The manuscript is well written and structured, the introduction provides sufficient background, and the cited references are relevant to the research. However, some changes are necessary before to publish. First, the title should be modified to explain that the function and phylogenetic analysis is a theoretical analysis, since the authors have only carried out an expression analysis. On the other hand, the authors must improve the material and methods section, since they have not explained how the gene expression analysis was carried out. They only indicate that the data have been obtained from previous unpublished studies from their laboratory, but they must explain how they obtained the samples, how they obtained the RNA, what method they used to carry out the gene expression analysis, and what analysis methods. statistics have used. In addition, and more specifically, other changes are necessary such as:

Response: Thank you very much for your valuable comments. In response to your suggestion, we have revised the title to: "Theoretical Analysis and Expression Profiling of 17β-Hydroxysteroid Dehydrogenase Genes in Gonadal Development and Steroidogenesis of Leopard Coral Grouper (Plectropomus leopardus)". This title more accurately reflects that our study combines theoretical analysis with gene expression profiling, emphasizing the scope and approach of our research.

Regarding materials and methods, make corresponding modifications according to your requirements. Supplemented with RNA extraction, differentially expressed gene analysis, transcriptome verification, etc., modified as follows:

For further validation, we extracted RNA from gonadal tissues using the TRIzol method, ensuring high quality (RIN > 7). For mRNA sequencing, we employed bead-based mRNA isolation and sequencing on the Illumina platform. Differential gene expression was analyzed with DESeq, focusing on genes with significant changes in expression (log2 fold change greater than 1, p-value less than 0.05). To validate these findings, selected differentially expressed genes (DEGs) were further examined using qPCR on the Applied Biosystems 6300 RT-PCR system. During qPCR analysis, the β2-microglobulin (b2m) gene was used as an internal control, and the relative expression levels were quantified using the 2-△△Ct method, ensuring the reliability of the transcriptomic data. Line:416-425.

  1. Lines 64, 81 and 92: eliminate “etc”. If other genes exist, the author should enumerate it and cited references.

Response: Thanks for your suggestion, we have corrected this error. At the same time, the corresponding genes were supplemented.

  1. Line 84: the authors should refer to the gene family?

Response: Thank you for your review and valuable suggestions on our paper. We have revised this section by adding information about the gene family to which the gene is mentioned. Modify as follows:

Among these, 17β-Hsd, also known as Hsd17b, is a key enzyme in the synthesis of sex steroid hormones, acting downstream in the sex hormone synthesis signaling pathway. It possesses both oxidative and reductive activities and is primarily responsible for the mutual conversion between estrone (E1), E2, androstenedione (A), and T. To date, 15 types of HSD17Bs (Hydroxysteroid (17-beta) Dehydrogenases) have been identified in mammals, whereas in fish, the variety of these enzymes is relatively less. Currently, only 11 types of Hsd17b family genes have been reported in fish. In existing research on fish, Kazeto et al. successfully transfected the Hsd17b1 gene from the Jap-anese eel into HEK 293 cells, where they observed its capability to convert E1 into E2. Furthermore, Rajakumar et al. discovered that during the gonadal development and maturation process in the walking catfish (C. batrachus), Hsd17b1 exhibited high ex-pression levels in the ovaries, suggesting a significant role for Hsd17b1 in regulating sex hormone levels during gonadal development and gametogenesis. Line:77-89.

  1. Line 369: specify the method and quantities used to anesthetize the animals.

Response: Thank you for your review and valuable suggestions on our paper. We have modified this part. Modify as follows: Prior to sampling, the fish were anesthetized using eugenol at a concentration of 30 mg/L, and 15 individuals were selected at each stage for gonad removal. Line:375-377

  1. Line 372: “in situ” in italics.

Response: Thanks for your suggestion, we have corrected this error.

  1. Section “Statistical analysis”: the authors should check the normality and homoscedasticity of data before to use parametric statistical test.

Response: Thank you for your review and valuable suggestions on our paper. We have modified this part. Modify as follows: Statistical analysis was performed using GraphPad Prism 9 (GraphPad Software, Inc., California, USA). Before applying parametric statistical tests, we checked the normality and homoscedasticity of the data to ensure that the underlying assumptions were met. Line:444-446

Round 2

Reviewer 2 Report

Comments and Suggestions for Authors

None